# Relationship between Partner Phubbing and Parent–Adolescent Relationship Quality: A Family-Based Study

**DOI:** 10.3390/ijerph20010304

**Published:** 2022-12-25

**Authors:** Julan Xie, Ya Luo, Zhuo Chen

**Affiliations:** School of Business, Central South University, Changsha 410083, China

**Keywords:** partner phubbing, parent–adolescent relationship quality, spillover effect, compensatory effect, dyadic model

## Abstract

A high-quality parent–child relationship is critical to the health and well-being of adolescents and, in the family system, the interaction pattern within couples is a decisive factor in parent–adolescent relationship quality. Using dyadic data from 441 Chinese couples, in this study, we examined the association between partner phubbing (a negative interaction behavior initiated by the spouse) and parent–adolescent relationship quality, and further explored the moderating effect of adolescent gender. Dyadic modeling showed that partner phubbing had both an intra-person effect and an inter-person effect on parent–adolescent relationship quality. For the intra-person effect, husbands’ phubbing had an adverse effect on the mother–adolescent relationship quality, and this effect was stronger for girls than boys; wives’ phubbing had a positive effect on the father–adolescent relationship quality, but this effect was only significant for boys. For the inter-person effect, the negative influence of husbands’ phubbing on father–adolescent relationship quality was only significant for boys; wives’ phubbing was uncorrelated with mother–adolescent relationship quality. These findings deepen our understanding of the links between the marital subsystem and the parent-adolescent subsystem in the family, underscore the importance of positive marital interactions for adolescent development, and have implications for personal smartphone use management in family contexts.

## 1. Introduction

Mobile technology such as smartphones has made social connection very convenient, but the problems they create are non-negligible [1,2]. One prominent issue is phubbing. The term “phubbing” is a portmanteau of “phone” and “snubbing”, which describes the act of someone being preoccupied with their smartphones and ignoring their partners during face-to-face communication [3,4]. Globally, mobile Internet users spend an average of 4.2 h on their smartphones per day, and digital life habits are being deeply cultivated [5]. With the penetration of smartphones, phubbing has been captured in various social situations, especially in the family environment [6,7,8].

Adolescence is an essential period of growth and transformation, and parents play a significant protective role in the adaptation and development of adolescents [9]. In family systems, the quality of the parent–adolescent relationship can greatly affect an adolescent’s health and well-being [10]. Several studies have shown that a high quality parent–adolescent relationship is beneficial for children’s positive and healthy development, including coping skills, emotional regulation, value formation, and socialization [11,12,13]. By contrast, a low quality parent–adolescent relationship is associated with children’s low well-being, poor mental health, and unhealthy behaviors [14,15].

The parent–adolescent relationship can be influenced by various factors within a family, including marital interactions [16,17]. A conflictual marital relationship has been shown to be associated with lower parental attention and sensitivity, and greater parental expressions of negative emotions toward their children [18,19,20]. Therefore, the interaction patterns and behaviors within couples are thought to be an important determinant of parent–adolescent relationship quality.

Partner phubbing, as a negative interaction behavior initiated by an intimate partner or spouse, has attracted extensive academic attention for its possible adverse relational and health consequences [19,21,22]. Specifically, partner phubbing has been shown to undermine conversation intimacy [23], exacerbate conflicts [24], decrease overall relationship satisfaction [19,25,26], and increase the mental health concerns of the partner who is phubbed [21,27,28]. Despite ample evidence that partner phubbing has adverse influences in the family environment, there is a lack of consideration for the association between partner phubbing and parent–adolescent relationship quality.

In family systems, all family members are interdependent, and family subsystems (e.g., marital and parent-child subsystems) are closely connected [29,30]. Previous research has well-established the negative effects of partner phubbing on the marital relationship [16,24,28], and thus, it is reasonable to assume that the adverse influences of phubbing might extend beyond the marital subsystem to the parent-adolescent subsystem. Indeed, there is evidence that a parent’s problematic use of technology could decrease co-parenting quality and also impede high-quality parent–child interactions [31]. Surprisingly, little attention has been paid to the potential influences of partner phubbing on the parent–adolescent relationship.

The spillover hypothesis [16] may help to clarify the expected correlation between partner phubbing and parent–adolescent relationship quality. Spillover refers to a sociological transfer of behavior and emotion from one setting to another in the same direction [16]. One example would be the negative affect or conflict that was generated in the marital dyad being expressed in the parent–child dyad. Based on the spillover hypothesis, marital relationship quality has been shown to have a significant impact on parent–child relationship quality [32,33].

Within the framework of the spillover hypothesis [16], in this study, we propose and test a dyadic model in which partner phubbing has negative intra-person and inter-person effects on parent–adolescent relationship quality, and these associations vary by adolescent gender. In this study, wives’ perceived partner phubbing was simplified as husbands’ phubbing; husband’s perceived partner phubbing was expressed as wives’ phubbing. The husband-wife matched data were collected from Chinese couples. The hypothesized model was tested using structural equation modeling. This research has the potential to advance existing knowledge of the determinants of the parent–adolescent relationship and to provide guidance for personal smartphone use in family contexts. Our work also has practical value for improving the well-being and healthy development of adolescents.

### 1.1. Partner Phubbing and Parent–Adolescent Relationship Quality

Partner phubbing might be a threat to the marital or co-parenting relationship. It has been found that partner phubbing could reduce marital relationship satisfaction [16,19] and undermine co-parenting quality [31,34]. Meanwhile, marital quality can significantly affect the parent–adolescent relationship [35,36]. In a family system, phubbing would hinder couples’ interactions and reduce coordination among family members, especially during co-parenting [31]. When there is poor co-parenting, parent–adolescent interactions could be undermined [37]. In line with this logic, partner phubbing might not only affect the marital relationship, but might also interfere with parent–adolescent interactions.

The intra-person effect is an important form of spillover in which one’s precondition in one setting affects his or her own outcomes in another setting [16]. For example, wives’ perceived marital relationship satisfaction (marital dyad) has been shown to have a positive correlation with their reports of the quality of the parent–child relationship (parent–child dyad), and the same results also hold for husbands [33]. A daily diary study has also shown that wives’ and husbands’ who perceived marital quality (marital dyad) both had a positive association with their perceived parent–child relationship quality (parent–child dyad) [36].

Partner phubbing may have an intra-person effect on parent–adolescent relationship quality. Previous research has shown that partner phubbing may spark marital conflict [24] and lead to negative moods in those who are phubbed [38]. It has been found that family conflicts between spouses may spillover to the parent–adolescent relationship, and the parents’ negative moods serve as a transferring mechanism [39]. During parent–adolescent interactions, negative emotions generated in unpleasant marital interactions may be displaced onto children, thus, undermining positive parent–child interactions and inducing a decline in parent–adolescent relationship quality. Based on the above evidence, we put forward the following hypotheses:

**Hypothesis** **1** **(H1a).***Husbands’ phubbing is negatively correlated with mother–adolescent relationship quality*.

**Hypothesis** **1** **(H1b).**
*Wives’*
*phubbing is negatively correlated with father–adolescent relationship quality.*


Different from the intra-person effect, the inter-person effect represents the effects of a person’s characteristics or behaviors on another person’s outcome [40]. In addition, the inter-person effect of the spillover hypothesis implies that one’s precondition in one setting affects his or her partner’s outcomes in another setting [16]. For instance, prior work showed a positive correlation between wives’ reports of relationship satisfaction (marital dyad) and husbands’ reports of co-parenting cooperation (parent–child dyad), whereas wives’ perceived negative couple interactions (marital dyad) were positively associated with husbands’ reports of hostile parenting (parent–child dyad) [41]. Research has also found that one spouse’s daily parenting stress (parent–child dyad) may be positively correlated with the other spouse’s perceived negative couple interactions (marital dyad) [42].

Partner phubbing may also have an inter-person effect on parent–adolescent relationship quality. That is, a spouse’s perceived partner phubbing in marital subsystem may be correlated with another spouse’s perception of the parent–adolescent relationship quality in the parent-child subsystem. After being phubbed by their spouse, parents may pour out their unhappiness to their children [43]. There is a strong emotional connection between parents and children [44], and children are able to empathize with their parents’ depressed moods [45]. They may even feel resentment toward the phubber for causing the other parent to experience negative emotions [46]. As a result, one spouse’s perceived partner phubbing might undermine the quality of the relationship between the phubber and the adolescent.

In addition, previous research has suggested that one parent’s experiences and behaviors were associated with the quality of communication between the other parent and the adolescents [8]. Therefore, one spouse’s negative experience in the marital subsystem, such as perceived partner phubbing, may affect the other spouse’s communication with their children in the parent-child subsystem. Furthermore, parents and children stay together during most family time. When a spouse perceives partner phubbing (marital dyad), their children may also perceive phubbing from one parent, which may directly impair the relationship quality between the phubber and the adolescents (parent–child dyad) [6]. Building on the spillover hypothesis and the empirical evidence discussed above, we put forward the following hypotheses:

**Hypothesis** **2** **(H2a).**
*Husbands’*
*phubbing is negatively correlated with father–adolescent relationship quality.*


**Hypothesis** **2** **(H2b).**
*Wives’*
*phubbing is negatively correlated with mother–adolescent relationship quality.*


### 1.2. Gender Differences among Adolescents

Gender differences in parent–adolescent relationships [8,47,48] may have implications for conceptualizing possible sex differences in the association between partner phubbing and parent–adolescent relationship quality. As for the intra-person effect, previous research has suggested that adolescents develop their own sex-typical personality through learning from their same-sex parents [49], and this process shapes a close link in same-sex parent–adolescent relationships. Similarly, prior work has shown that parents have stronger bonds with children of the same sex [48], suggesting that children are more likely to be influenced by the same-sex parent. Therefore, the intra-person effect of partner phubbing on parent–adolescent relationship quality might vary by adolescent gender.

Research has suggested that girls learn feminine characteristics from their mother [50], and girls may have more frequent interactions with their mothers. Mothers have been shown to talk more to their daughters about unpleasant marital experiences than to their sons [51]. Therefore, girls are more susceptible to their mothers’ moods during their interactions [52]. In line with this notion, mothers’ negative emotions induced by fathers’ phubbing may have a greater impact on the mother–daughter relationship quality than on the mother–son relationship quality.

In contrast, sons develop their masculine personality mainly through learning from their fathers [53], and they have more interactions with their fathers. Husbands may be more likely to express negative moods induced by wives’ phubbing with their sons than with their daughters. Therefore, wives’ phubbing may have a greater impact on the father–son relationship quality than on the father–daughter relationship quality. Meanwhile, boys are not as considerate as girls [52,54], and they may not offer consolation when their fathers are phubbed. The negative affections fathers experienced during marital interactions are more likely to hinder positive father–son interactions. Overall, the discussion above leads to the following hypotheses regarding moderation of the intra-person effect:

**Hypothesis** **3** **(H3a).**
*The effect of husbands’ phubbing is greater on mother–daughter relationship quality than on mother–son relationship quality.*


**Hypothesis** **3** **(H3b).**
*The effect of wives’ phubbing is greater on father–son relationship quality than on father–daughter relationship quality.*


The inter-person effect of partner phubbing on parent–adolescent relationship quality may also vary by adolescent gender. In the family system, daughters usually have an alliance with their mothers [54], and they are more sympathetic to their mothers’ feelings [55]. In this condition, daughters will be on the same side as their mothers and less willing to be close to their fathers [52]. Research has also shown that father–daughter relationship quality is not as high as father–son relationship quality [56]. Thus, husbands’ phubbing may have a greater impact on father–daughter relationship quality than on father–son relationship quality.

By the same token, the father is an important model for boys to develop masculinity [53]. Therefore, sons can be expected to have more respect for their fathers and to more easily develop a close relationship with them [48,57]. In addition, boys may avoid forming a close relationship with their mothers because they develop a different gender identity from that of their mothers [54]. Husbands may also be more likely to share with their sons than their daughters their negative moods generated by partner phubbing. This may lead sons to distance themselves from their mothers, thus, undermining the mother–son relationship quality. Taken together, with regard to the gender differences in the inter-person effect, we put forward the following hypotheses:

**Hypothesis** **4** **(H4a).**
*The effect of husbands’ phubbing is greater on father–daughter relationship quality than on father–son relationship quality.*


**Hypothesis** **4** **(H4b).***The effect of wives’ phubbing is greater on mother–son relationship quality than on mother–daughter relationship quality*.

### 1.3. The Present Study

Drawing upon the spillover hypothesis [16], this study developed a conceptual model of the intra-person effects and inter-person effects of partner phubbing on parent–adolescent relationship quality; the conceptual model also included adolescent gender as a moderator of these effects. Husband-wife matched data were collected from 441 Chinese couples. We used the dyadic data analysis method to test the hypotheses. The results should enrich the existing literature on the antecedents of parent–adolescent relationship quality and the consequences of partner phubbing in the family system. Our study may also have practical implications for parents to promote adolescent health and development by managing personal daily technology use.

## 2. Materials and Methods

### 2.1. Participants

Chines married couples (*N* = 507) were recruited using convenience sampling. A total of 441 couples gave valid reports of partner phubbing, parent–adolescent relationship quality, and demographic information, with a valid response rate of 86.98%. The mean age of fathers was 42.27 years (range = 32–60, SD = 4.15) and the mean age of mothers was 39.70 years (range = 30–56, SD = 4.01); 37.4% of mothers and 48.1% of fathers had a degree higher than college. The average marriage length was 16.18 years (range = 10–33, SD = 3.71). For adolescents, 249 adolescents were boys and 192 were girls. The mean age of adolescents was 12.97 years (range = 11–15, SD = 0.74). 

### 2.2. Measures

#### 2.2.1. Partner Phubbing

Partner phubbing was measured using a 9-item scale developed by Roberts and David [27]. An example is “My partner glances at his/her cell phone when talking to me.” Participants rated each item on a five-point scale (from 1 = never to 5 = always). Responses to all items were averaged, with higher scores indicating a higher level of perceived partner phubbing. This instrument has shown good validity and reliability among Chinese samples [22,28]. The Cronbach’s alphas were 0.77 and 0.80 for husbands and wives, respectively.

#### 2.2.2. Parent–Adolescent Relationship Quality

Parent–adolescent relationship quality was assessed with the Middle School Student’s Parent–Child Relationship Questionnaire [58]. This 26-item questionnaire is designed for parents to assess their relationship with their children, and consists of four dimensions. The dimension “understanding and communication” refers to the aspect that parents fully understand and support their children, and they can communicate unimpeded; an example item is “I know what my child likes and dislikes.” The dimension “excoriation and controlling” refers to the parents’ attempts to control their children by their own subjective will, and this dimension is reverse scored; an example item is “My conversations with my child are imperative or interrogative.” The dimension “liking and esteem” refers to the parents affection and respect for their children; an example item is “I like being with my child.” The dimension “growth and tolerance” refers to the aspect that parents care about their children’s growth and have a tolerant attitude towards their children’s words and deeds during their growth; an example item is “I am tolerant of my child’s failure in trying something.” Parents rated each item on a five-point scale (from 1 = not true at all to 5 = completely true). A higher average score reflected higher parent–adolescent relationship quality. The Cronbach’s alphas were 0.86 and 0.88 for fathers and mothers, respectively.

#### 2.2.3. Control Variable

Previous research has found that the negative association between partner phubbing and relationship satisfaction was significant among couples married for more than seven years [28]. Considering that the couples in the sample had been married for more than 10 years, we included length of marriage as a control variable.

### 2.3. Procedure

The current research used questionnaires to collect matched husband-wife data. All research procedures were in accordance with the ethical standards of the researchers’ University Ethics Committee and with the 1964 Helsinki Declaration and its later amendments. We recruited adolescents from three middle schools in Mainland China to participate in our investigation. Under our guidance, each adolescent took home two envelopes, one for each parent. Each envelope contained an invitation letter with information about the study, including that their participation was voluntary, their answers were anonymous, and their child would not be penalized if one or both parents decided not to participate. Each envelope also included a set of questionnaires, which parents were asked to complete separately. Each parent was asked to seal the completed questionnaires in an envelope provided by the researchers, without their answers being seen by their spouse. There was no identifying information on the questionnaires or on the envelopes. The next day, the adolescents took the two sealed envelopes back to school and handed them over to our researchers. All students received a small gift for their help with the study.

### 2.4. Data Analysis

Descriptive analyses and paired *t*-tests were conducted in SPSS 21.0 (IMB, New York, NY, USA). The dyadic data analysis method was used given the interdependence of matched husband-wife data [59]. Structural equation modeling (SEM) was employed using AMOS 24.0 (IMB, New York, NY, USA) to examine the dyadic model. Missing data were handled using the expectation maximization algorithm [60].

In the dyadic model, we correlated the errors of the same indicators reported by husbands and wives. In addition, we made the residuals of husbands’ phubbing and wives’ phubbing to be correlated, and the residuals of mother–adolescent relationship quality and father–adolescent relationship quality to be correlated. We used parcels as the indicators for each latent variable. Specifically, a factorial algorithm [61] was used to create parcels of items for latent variable partner phubbing, and divided the items into three parcels. For latent variable parent–adolescent relationship quality, we used the four dimensions of the scale to be the parcels.

## 3. Results

### 3.1. Preliminary Analysis

Means, standard deviations, and Pearson correlation coefficients for all variables are shown in Table 1. Wives’ phubbing was negatively correlated with father–adolescent relationship quality, but was not significantly correlated with mother–adolescent relationship quality. In addition, husbands’ phubbing was negatively correlated with mother–adolescent relationship quality and with father–adolescent relationship quality. The paired *t*-tests showed that the average partner phubbing score was higher for wives (M = 2.95, SD = 0.61) than for husbands (M = 2.86, SD = 0.57), *t*(440) = 3.08, *p* = 0.002. 

### 3.2. Dyadic Model

We conducted a test of the measurement model before conducting the dyadic analysis. The measurement model consisted of four latent variables: husbands’ phubbing, wives’ phubbing, mother–adolescent relationship quality, and father–adolescent relationship quality. The measurement model showed a good fit to the data (χ^2^_(65)_ = 146.58, χ^2^/df = 2.26, IFI = 0.97, CFI = 0.97, TLI = 0.95, and RMSEA = 0.053). 

The dyadic analysis was conducted to test the association between partner phubbing and parent–adolescent relationship quality. The results indicated a good fit to the data (χ^2^_(74)_ = 119.87, χ^2^/df = 1.62, IFI = 0.98, CFI = 0.98, TLI = 0.97, and RMSEA = 0.038). As represented in Figure 1, husbands’ phubbing was negatively associated with mother–adolescent relationship quality (*B* = −0.21, *p* < 0.001) and father–adolescent relationship quality (*B* = −0.14, *p* < 0.01). Therefore, H1a and H2a were supported. Meanwhile, wives’ phubbing was positively associated with father–adolescent relationship quality (*B* = 0.14, *p* < 0.01), and was not associated with mother–adolescent relationship quality (*B* = 0.07, *p* > 0.05), thus, failing to support H1b and H2b. Overall, the intra-person and inter-person spillover effect hypotheses were supported for wives, but not for husbands.

We further tested the moderating effect of adolescent gender. First, we established a multi-group model that included a boy dyadic model and a girl dyadic model, and then compared the corresponding paths. The value of critical ratios for differences between parameters was used to test the difference between two groups. A moderating effect exists when the absolute value is more than 1.96 (*p* < 0.05) [62,63]. The results of tests of the dyadic models are presented in Figure 2 (for boys) and Figure 3 (for girls). The results indicated a good fit both for the boy dyadic model (χ^2^_(74)_ = 95.46, χ^2^/df = 1.29, IFI = 0.99, CFI = 0.98, TLI = 0.98, and RMSEA = 0.034), and for the the girl dyadic model, (χ^2^_(74)_ = 83.56, χ^2^/df = 1.13, IFI = 0.99, CFI = 0.99, TLI = 0.99, and RMSEA = 0.026).

For the intra-person effect, husbands’ phubbing was negatively associated with both mother–son relationship quality (*B* = −0.12, *p* < 0.05) and mother–daughter relationship quality (*B* = −0.32, *p* < 0.001). Because the association between husbands’ phubbing and mother–adolescent relationship quality was significant in both models, the multi-group analysis was then conducted in AMOS 24.0 to examine differences based on adolescent gender. We compared the path parameter (husbands’ phubbing → mother–adolescent relationship quality) in the boy dyadic model and the girl dyadic model. The value of critical ratios for differences between parameters was −2.02, and the absolute value was greater than 1.96, suggesting that the correlation between husbands’ phubbing and mother–adolescent relationship quality was significantly stronger for girls than for boys. In addition, wives’ phubbing had a significant positive effect on father–son relationship quality (*B* = 0.13, *p* < 0.01), but was not significantly correlated with father–daughter relationship quality (*B* = 0.09, *p* > 0.05).

In terms of the inter-person effect, husbands’ phubbing was negatively correlated with father–son relationship quality (*B* = −0.15, *p* < 0.001), but was not significantly correlated with father–daughter relationship quality (*B* = −0.12, *p* > 0.05). Inconsistent with our hypotheses, wives’ phubbing was not correlated with either mother–son relationship quality (*B* = 0.07, *p* > 0.05) or with mother–daughter relationship quality (*B* = 0.10, *p* > 0.05).

## 4. Discussion

The present research used a cross-dyadic design to examine the association between partner phubbing and parent–adolescent relationship quality in a family system. Our study confirmed that partner phubbing had intra-person and inter-person effects on parent–adolescent relationship quality. In addition to the spillover effect, we also discovered a compensatory effect of the marital system on the parent-child system. Furthermore, adolescent gender moderated this association. This study extends previous research on the antecedents of parent–adolescent relationship quality, and on the consequences of partner phubbing. The results have potential applied value for promoting adolescent health and well-being, improving family interaction patterns, and managing personal technology use in family systems.

### 4.1. The Intra-Person Effect of Partner Phubbing on Parent–Adolescent Relationship Quality

In terms of the intra-person effect, the results indicated that husbands’ phubbing was negatively correlated with mother–adolescent relationship quality. Prior research has shown that women tend to be more vulnerable to suffer from life stress than men [64,65]. Phubbing can be regarded as a life stress that could threaten the psychological needs of the partner being phubbed and lead to negative moods [3], and ultimately damage marital relationship quality [4,27]. Parents under stress (in this case from marriage) are less able to be emotionally sensitive to and respond to the needs and desires of their children [66]. The negative emotions induced by partner phubbing that wives experience during marital interactions may spill over into mother–adolescent interactions, further impeding mother–adolescent relationship quality. Therefore, husbands’ phubbing may have a negative effect on mother–adolescent relationship quality.

The results also showed that husbands’ phubbing had a greater effect on mother–daughter relationship quality than on mother–son relationship quality. The mother–daughter relationship is a same-sex dyad in which mother and daughter both hold a feminine viewpoint, so that they have a shared knowledge [67]. Girls have been shown to provide more interpersonal care than boys, and mother–daughter dyads might develop more mutual concerned responsiveness than mother–son dyads [55]. Given that daughters generally interact more with their mothers than sons do [45], wives’ negative emotions generated from partner phubbing would be more likely to be transmitted to daughters than to sons. There has been evidence reported that the relationship between maternal depression and daughter depression is stronger than the relationship between maternal depression and son depression [68]. Therefore, husbands’ phubbing would have a greater influence on mother–daughter relationship quality than on mother–son relationship quality.

Meanwhile, our study revealed that wives’ phubbing was positively related to father–adolescent relationship quality. The compensatory hypothesis [16] may provide a reasonable explanation for the results, which refers to a process by which a person seeks opposite experiences and satisfactions in one system to compensate or make up for deficiencies in another system. According to the compensatory hypothesis [16], parents who cannot meet their needs for attention or intimacy in the marital relationship would seek to satisfy these needs in the parent–adolescent relationship. In addition, men often have better adaptability than women when facing life stress [64,65]. Therefore, husbands might compensate for unsatisfactory marital relationships by increasing involvement with and investment in their children, thus, improving the father–adolescent relationship quality. Taken together, these lines of reasoning suggest that wives’ phubbing may be beneficial to the parent–adolescent relationship.

However, wives’ phubbing had a positive association with father–son relationship quality, but not with father–daughter relationship quality. The father–son relationship is another same-sex dyad. Holding the same male viewpoint, father and son have a shared knowledge [67]. As a means of compensation, fathers who experienced partner phubbing in the marital relationship may prefer to communicate with sons than daughters to restore positive emotions and meet the psychological needs of relatedness and intimacy. By contrast, a sense of distance and difference is often characteristic of father–daughter relationships [69]. Fathers communicate less with their daughters than with their sons [70], and are especially less likely to talk about unhappy marital experiences with their daughters. Therefore, father–daughter relationship quality may not be affected by wives’ phubbing.

### 4.2. The Inter-Person Effect of Partner Phubbing on Parent–Adolescent Relationship Quality

In terms of the inter-person effect, the results indicated that husbands’ phubbing was negatively related to father–adolescent relationship quality. As aforementioned, men adapt to stress better than women [64], suggesting that wives have a lower threshold for becoming upset when experiencing partner phubbing. Research has shown that adolescents perceived better mother–adolescent relationships than father–adolescent relationships [71]. In a family system, the adolescents tend to fight one parent against the other to form a stable alliance [46]. Mothers generally spend more time building closer relationships with their children, and therefore, children may be more likely to become part of a mother–child alliance, and they might back their mothers up by distancing themselves from their fathers. Thus, husbands’ phubbing could have a negative effect on father–adolescent relationship quality.

We found that the correlation between husbands’ phubbing and father–son relationship quality was stronger than that between husbands’ phubbing and father–daughter relationship quality. Sons may feel obligated to protect their mothers as an expression of their hegemonic masculinity [72]. Research has also shown that the increase in parent–adolescent conflict during adolescence is steeper for boys than for girls [73]. In line with this logic, sons are more likely to be hostile to their fathers than daughters when they learn that their mothers have suffered from their fathers’ phubbing behaviors. Therefore, husbands’ phubbing may have a greater effect on father–son relationship quality than on father–daughter relationship quality.

Furthermore, our results indicated that wives’ phubbing was not significantly associated with mother–adolescent relationship quality. This may be because the male masculine personality would make husbands better able to maintain emotional boundaries and limit the spillover of negative emotions caused by partner phubbing to parent–adolescent interactions [17]. In addition, adolescents often have a better relationship quality with their mothers than their fathers [74]. As a result, mother–adolescent relationship quality may be less likely to be affected by wives’ phubbing.

### 4.3. Theoretical Contributions

This study makes several theoretical contributions to the literature. First, parent–adolescent relationship quality has been shown to play a critical role in adolescents’ health and well-being [9], and numerous studies have investigated the protective factors of parent–child relationship quality in family environment [12,31,75]. However, previous research has mainly focused on a single subsystem (i.e., the parent-child subsystem). The current study broadened the research scope to the whole family system by investigating the potential influences of partner phubbing on parent–adolescent relationship quality. Our study further supported the spillover hypothesis between the marital dyad and the parent–child dyad. Thus, the findings highlight the importance of positive marital interactions for adolescents’ development and well-being.

Second, previous research has mainly focused on the effects of partner phubbing on the partner who experienced the phubbing at the individual level [4,7,76]. Our research complemented previous work by distinguishing the inter-person and intra-person effects of partner phubbing on parent–adolescent relationship quality at the dyadic level, and contributed to a more comprehensive understanding of the negative influences of phubbing on family systems. This study provides a new research paradigm for the parent–child relationship literature, and has value as a foundation for future research.

Third, the use of dyadic analyses allowed us to investigate the consequences of partner phubbing more systematically. Our results indicated that partner phubbing has spillover and compensatory effects on parent–adolescent relationship quality. Several studies have explored the “dark sides” of phubbing on interpersonal relationships and health issues [3,21,25], yet, no research has found the potential “bright sides” of phubbing. Our study added to the previous literature by revealing that wives’ phubbing can trigger a compensatory process and may be beneficial to parent–adolescent relationship quality.

Finally, the association between partner phubbing and parent–adolescent relationship quality was different for boys and girls. Specifically, the intra-person effect of husbands’ phubbing on mother–adolescent relationship quality was stronger for girls than boys, while the inter-person effect of husbands’ phubbing on father–adolescent relationship quality was only significant for boys. Furthermore, the compensatory effect of wives’ phubbing on father–adolescent relationship quality was only significant for boys. Our results offer insights about sex differences in parent–adolescent relationships, and contribute to a more comprehensive understanding of the possible boundary conditions of the partner phubbing effects in the family context.

### 4.4. Practical Implications

This study has significant practical implications for the development of positive family relationships. First, the negative consequences of husbands’ phubbing have spillover effects on mother–adolescent relationship quality and on father–adolescent relationship quality. Research has suggested that wives as caregivers and nurturers spend more time with their children than fathers do in most families [77]. Parent–adolescent relationship quality might be more likely to be affected by wives’ moods and states. Based on our results, wives are encouraged to establish an emotional boundary to reduce the spillover of negative emotions experienced in the marital relationship to the parent–adolescent relationship. We also found that mother–daughter relationship quality was more susceptible to husbands’ phubbing, and therefore, wives should avoid communicating with their daughters about negative experiences in the marital relationship. Meanwhile, reducing the incidence of husbands’ phubbing is necessary. Husbands are encouraged to avoid sacrificing family time for phone use to promote a healthy family atmosphere.

Furthermore, our findings suggest preventing the adverse effects of partner phubbing from spilling over into parent–child interactions by responding positively to phubbing issues. Specifically, wives and husbands should both use proactive approaches to respond to partner phubbing rather than immerse themselves in negative emotions. For example, they are encouraged to talk openly about how partner phubbing affects their marital interactions and come up with constructive solutions, so that it does not further negatively affect parent–child relationship quality.

### 4.5. Limitations and Future Directions

When interpreting our findings, several limitations should to be noticed. First, the present research was a cross-sectional study, which cannot provide evidence of causal relationships. Future studies could conduct cross-lagged analysis with a longitudinal design to test causal associations between partner phubbing and parent–adolescent relationship quality. Second, this study did not examine any mediation mechanism that would explain the association between partner phubbing and parent–adolescent relationship quality. Additional research is needed to explore the potential mediators to refine our findings. For example, a prior study found that marital conflict was a potential mediator between partner phubbing and co-parenting quality [31], which may have implications for understanding the effect of partner phubbing on parent–adolescent relationship quality.

In addition, the current study was conducted in a Chinese sample, which may limit generalization of the results. Living in a collectivistic culture, Chinese parents have stronger emotional dependence on their children and share more with them, however, in Western families characterized by individualism, parents and adolescents are relatively independent [78]. The effect of partner phubbing on parent–adolescent relationship quality detected in this study might be weaken when generalized to Western cultures. Future work on multicultural evaluations should be considered. Futhermore, the instrument for measuring phubbing in this study reflected the daily perceived partner phubbing behavior, but could not measure the specific reasons and motives behind phubbing. Futute research can measure phubbing in more specific situations in which the relationship between partners and the relationship between parents and children are taken into consideration. Last, parent–adolescent relationship quality in this study was reported by parents. Considering that parents might show more bias than adolescents do in self-reports about parent–adolescent relationship quality, the measures of adolescents’ perceived parent–adolescent relationship quality could be used in future studies.

## 5. Conclusions

Based on the spillover hypothesis, this study examined the correlation between partner phubbing and parent–adolescent relationship quality. Our results revealed that partner phubbing had intra-person and inter-person effects on parent–adolescent relationship quality, and this association was moderated by the gender of adolescent. For the intra-person effect, husbands’ phubbing had a negative impact on mother–adolescent relationship quality, and this effect was stronger for girls than boys. By contrast, wives’ phubbing had a positive effect on father–son relationship quality, but no effect on father–daughter relationship quality. For the inter-person effect, husbands’ phubbing showed a negative effect on father–son relationship quality, but not on father–daughter relationship quality. Wives’ phubbing had no significant effect on mother–adolescent relationship quality. This study provides a comprehensive understanding of the association between partner phubbing and parent–child relationship. The results contribute to the existing literature on parent–adolescent relationship quality and phubbing, and have implications for the promotion of adolescent health development and personal management of smartphone use in family contexts.

## Figures and Tables

**Figure 1 ijerph-20-00304-f001:**
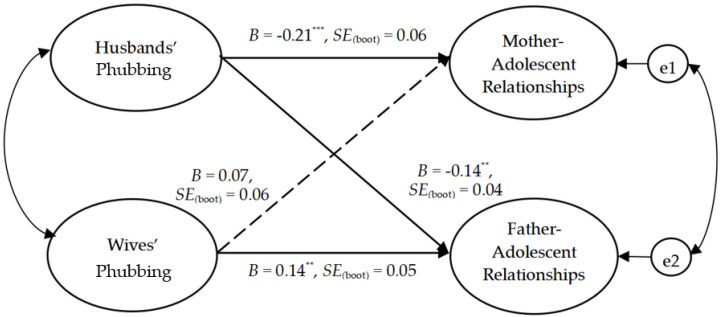
The dyadic model tested in the full sample. Note. *N* = 441 couples. ** *p* < 0.01, *** *p* < 0.001.

**Figure 2 ijerph-20-00304-f002:**
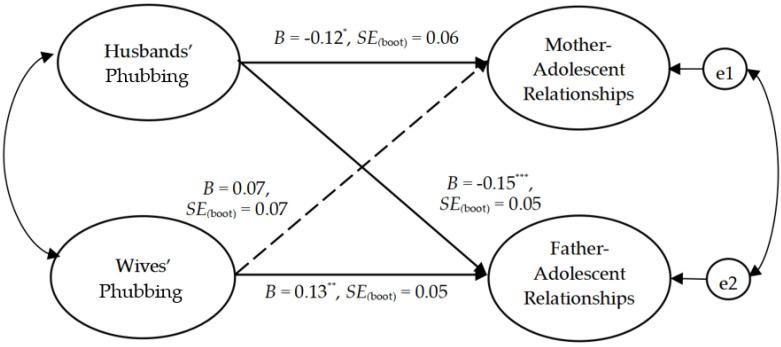
The dyadic model of the whole participants in the boy sample. Note. *N* = 249 couples. * *p* < 0.05, ** *p* < 0.01, *** *p* < 0.001. Dotted line indicates non-significant.

**Figure 3 ijerph-20-00304-f003:**
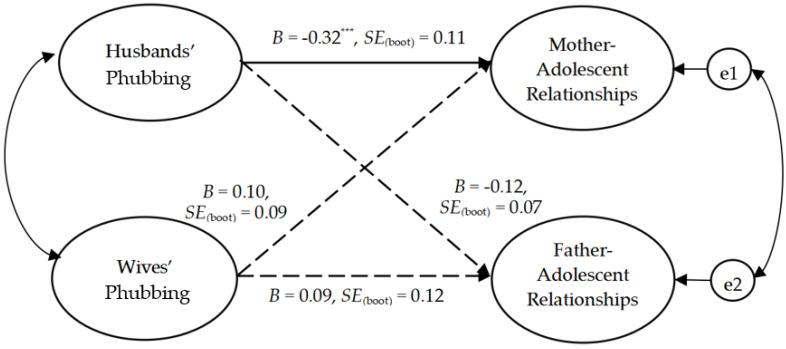
The dyadic model of the whole participants in the girl sample. Note. *N* = 192 couples. *** *p* < 0.001. Dotted line indicates non-significant.

**Table 1 ijerph-20-00304-t001:** Means, standard deviations, and correlation coefficients among the study variables.

Variables	M	SD	1	2	3	4	5
1. Marital length	16.18	3.71	-				
2. Wives’ phubbing	2.86	0.57	−0.15 **	-			
3. Husbands’ phubbing	2.95	0.61	−0.14 **	0.45 ***	-		
4. Father–adolescent relationship quality	3.63	0.43	0.10 *	−0.10 *	−0.18 ***	-	
5. Mother–adolescent relationship quality	3.82	0.44	−0.009	−0.08	−0.23 ***	0.40 ***	-

Note. *N* = 441 couples, * *p* < 0.05, ** *p* < 0.01, *** *p* < 0.001.

## Data Availability

The data presented in this study are available on request from the corresponding author. The data are not publicly available due to data privacy.

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
