# Peer review of "Relationship between Partner Phubbing and Parent–Adolescent Relationship Quality: A Family-Based Study"

_ijerph, 2022, doi:10.3390/ijerph20010304_

Round 1

Reviewer 1 Report

The research investigates the interaction of marital and parent-adolescent subsystem in the overall family structure. Specifically the study focus on how phubbing ( the action of neglecting one's physical interlocutor to frequently, more or less compulsively consult one's cell phone or other interactive device) affects familiar relationships, both intra-person ( how mothers’ phubbing conditions mothers-children relationship and how fathers’ phubbing conditions fathers-children relationship) and inter-person (how mothers’ phubbing conditions fathers-children relationship and how fathers’ phubbing conditions mothers-children relationship).

We consider the topic original and relevant in the field for several reasons: the lack of scientific literature that has previously deepened this theme, especially taking a specific point of view on the relationship binding every element of the familiar system; the use of smartphone is a contemporary factor having recently overturned our daily life, so we can consider it something on which focus our attention and study to well understand how our relationships are evolved, how they could keep changing and how we can impact on this, giving a psychodynamic and relational point of view to scientific protocol.

For the relevance that we attribute to the topic and for the potential that we see in this research to lead the way to additional insights, we believe some expedients could give greater emphasis to this work and help his comprehension. For example we suggest to switch the expression “wives’ partner phubbing” with “husband's phubbing” and so “husbands' partner phubbing” with “wive's phubbing”; for the same reason, we suggest to use a more immediate and easier expression to indicate how a phubber mother conditions her relationship with the son and how a phubbing father conditions his relationship with the children (intra-person effect). We also suggest to clarify what you mean with the expression “inter person effect”. Another suggestion could be to directly interview children about their relationships with parents instead of exploiting parents’ only point of view.

Although the chance to improve features concerning methodology, we found the references are appropriate and tables and figures really useful to make more intuitive research results, besides the statistical analysis is well compelling.

Finally we estimate that the conclusions are consistent with the evidence and arguments presented and they addressed the main question posed, even if it could be interesting and a subject for further researches, to evaluate the possible underlying factors of these evidences.

Author Response

Point 1: We suggest to switch the expression “wives’ partner phubbing” with “husband’s phubbing” and so “husbands’ partner phubbing” with “wive’s phubbing”; for the same reason, we suggest to use a more immediate and easier expression to indicate how a phubber mother conditions her relationship with the son and how a phubbing father conditions his relationship with the children (intra-person effect).

Response 1: We appreciate and accept your suggestion. In the revised paper, we have changed “wives’ partner phubbing” to “husbands’ phubbing” and changed “wives’ partner phubbing” to “husbands’ phubbing”. We also added a statement in the Introduction section that “In this study, wives’ perceived partner phubbing was simplified as husbands’ phubbing; while husband’s perceived partner phubbing was expressed as wives’ phubbing.” (page 2 lines 78-80 in red)

In addition, the association of our interest involves two family subsystems. Within each subsystem, the recipient of the phubbing behavior is different. For instance, in marital subsystem, husbands’ phubbing refers to the husbands’ smartphone usage and snubbing their wives; while in parent-child subsystem, fathers’ phubbing refers to the phone snubbing behavior initiated by fathers toward their children. Although the initiators of husbands’ phubbing and fathers’ phubbing are the same in the whole family system, the definitions of the two types of phubbing are essentially different. Therefore, we didn’t change “wives’ phubbing” to “mothers’ phubbing” nor change “husbands’ phubbing” to “fathers’ phubbing”.

Point 2: We also suggest to clarify what you mean with the expression “inter person effect”.

Response 2: Thank you for this comment. In the current edition, we modified the statement on the definition of the inter-person effect of the spillover hypothesis as follows: “Different from the intra-person effect, the inter-person effect represents the effects of a person’s characteristics or behaviors on another person’s outcome [1]. And the inter-person effect of the spillover hypothesis refers to the precondition in one setting which affects his or her partner’s outcomes in another setting [2]” (page 3 lines 117-120 in red). In addition, to eliminate the confusion in the expression “inter-person effect”, we further explained the inter-person effect of partner phubbing on parent-adolescent relationship quality, that is, a spouse’s perceived partner phubbing in marital subsystem may be correlated with another spouse’s perception of parent-adolescent relationship quality in parent-child subsystem (page 3 lines 129-131 in red).

Point 3: Another suggestion could be to directly interview children about their relationships with parents instead of exploiting parents’ only point of view.

Response 3: Thank you for raising this point. This study measured parent-adolescent relationship from the perspective of parents based on the following consideration. Our sample of adolescents were middle school students. Middle school is a period of transition to adulthood [3]. During this period, the adult sense and self-awareness of adolescents gradually enhanced due to physiological changes. At the same time, their thinking capabilities showed the characteristics of independence, criticism, and logic [4]. Compared with the physical and psychological upheaval of children in adolescence, the parents are relatively “stable”, and can report the parent-adolescent relationship more objectively [5].

We agreed with your suggestion and acknowledged that interviewing children directly about their relationships with parents is an effective way to measure parent-adolescent relationship quality, because adolescents might show less bias than parents in self-reports about parent-adolescent relationship quality. Therefore, in the Limitations and Future Directions section, we proposed that future studies can measure parent-adolescent relationship quality from the perspective of children. (page 12 lines 536-539 in red)

References:

  1. Cook,W. L.; Kenny, D.  The actor–partner interdependence model: A model of bidirectional effects in developmental studies[J]. Int. J. Behav. Dev. 2005, 29, 101–109.
  2. Erel, O.; Burman, B. Interrelatedness of marital relations and parent-child relations: A meta-analytic review. Bull.1995, 118, 108–132.
  3. Yeung,W. J. J.; Hu,  Coming of age in times of change: The transition to adulthood in China[J]. Annu. Am. Acad. Polit. Soc. Sci. 2013, 646, 149–171.
  4. Fuligni,A.  Authority, autonomy, and parent–adolescent conflict and cohesion: A study of adolescents from Mexican, Chinese, Filipino, and European backgrounds[J]. Devel psychol. 1998, 34, 782–792.
  5. Wu, J.; Guo, X.; Huang, X.; Li, S. The making of middle school student’s parent-child relationship questionnaire. Journal of Southwest University (Social Sciences Edition). 2011, 37, 39–44.

Reviewer 2 Report

A study that examined the association between partner phubbing and the quality of parent-adolescent relationships is reviewed in the manuscript Relationship between Partner Phubbing and Parent-Adolescent Relationship Quality: A Family-Based Study. Partner phubbing had both an intra-person and an inter-person effect on the quality of the parent-adolescent relationship.

The subject matter, the study and the line of thought are clearly presented in the manuscript, where an interesting and important contribution to understanding the links between the spouse’s use of smartphones and parent-adolescent relationships is provided.

However, below are a few suggestions for improvement.

It would help the reader if there was a definition of the term “phubbing” at the beginning of the manuscript.

L.    107-110

Hypothesis 1 (H1a). Mothers’ partner phubbing is negatively correlated with mother-adolescent relationship quality.

Hypothesis 1 (H1b). Fathers’ partner phubbing is negatively correlated with father-adolescent relationship quality.

In all hypotheses (H1a, H1b, H2a, H2b, H3a, H3b, H4a, H4b), use "mothers’", “wives’”, “fathers’” and "husbands" consistently.

L.    473

"This study also has significant practical implications for the development of positive family relationships."

Delete "also", because it is at the beginning of the section.

L 524-526

"By contrast, husbands’ partner phubbing had a compensatory effect on father-son relationship quality, but no effect on father-daughter relationship quality."

Why is this finding stated in relation to the interpretation of the findings rather than the direct findings of the study, like in the following two instances?

  1. 386-387

“Meanwhile, our study revealed that husbands’ partner phubbing was positively related to father-adolescent relationship quality.”

  1. 16-17.

“... husbands’ partner phubbing had a positive effect on the father-adolescent relationship quality, but this effect was only significant for boys.”

The data availability statement is missing and no link to the data set is provided.

See instructions: “In this section, please provide details regarding where data supporting reported results can be found, including links to publicly archived datasets analyzed or generated during the study. Please refer to suggested Data Availability Statements in section "MDPI Research Data Policies" at https://www.mdpi.com/ethics. If the study did not report any data, you might add "Not applicable" here.”

Author Response

Point 1: It would help the reader if there was a definition of the term “phubbing” at the beginning of the manuscript.

Response 1: We appreciate and accept this suggestion. In the current version, we clarified the definition of the term “phubbing” at the beginning of the manuscript (page 1 lines 30-32 in red).

Point 2: In all hypotheses (H1a, H1b, H2a, H2b, H3a, H3b, H4a, H4b), use “mothers’”, “wives’”, “fathers’” and “husbands’” consistently.

Response 2: Thank you for raising this point. In the current version, we used “wives’ phubbing” and “husbands’ phubbing” consistently in all hypotheses (lines 113-116, 141-151, 180-183, 201-204 in red).

Point 3: L. 473—”This study also has significant practical implications for the development of positive family relationships.” 

Delete “also”, because it is at the beginning of the section.

Response 3: We appreciate and adopt this advice. We deleted “also” in this sentence in the current version (page 11 lines 491-492 in red).

Point 4: L 524-526—"By contrast, husbands’ partner phubbing had a compensatory effect on father-son relationship quality, but no effect on father-daughter relationship quality."

Why is this finding stated in relation to the interpretation of the findings rather than the direct findings of the study, like in the following two instances?

L 386-387—“Meanwhile, our study revealed that husbands’ partner phubbing was positively related to father-adolescent relationship quality.”

L 16-17.—”... husbands’ partner phubbing had a positive effect on the father-adolescent relationship quality, but this effect was only significant for boys.”

Response 4: Thank you for this comment. In the revised paper, we modified the statement as follows: “By contrast, wives’ phubbing had a positive effect on father-son relationship quality, but no effect on father-daughter relationship quality.” (page 12 lines 547-548 in red)

Point 5: The data availability statement is missing and no link to the data set is provided.

Response 5: Thank you for pointing us to this issue. In the current reversion, we added the Data Availability Statement: The data presented in this study are available on request from the corresponding author. The data are not publicly available due to data privacy (page 12 lines 567-568 in red).

Reviewer 3 Report

The research deals with the relationship between partner phubbing and the relationship with adolescents. Although this is a topical and interesting phenomenon, I believe that there is a problem of a false relationship between the phenomenon and its consequences. Phubbing is a behaviour generated by higher order factors and can only play a mediating or moderating role between the quality of the relationship between partners and adolescents, but not be the cause. It is therefore necessary to have a theoretically grounded measure that focuses first and foremost on the relationship between partners and the relationship between parents and children, where phubbing is one of the phenomena that could improve the understanding of family relationships.

Author Response

Point 1: The research deals with the relationship between partner phubbing and the relationship with adolescents. Although this is a topical and interesting phenomenon, I believe that there is a problem of a false relationship between the phenomenon and its consequences. Phubbing is a behaviour generated by higher order factors and can only play a mediating or moderating role between the quality of the relationship between partners and adolescents, but not be the cause. It is therefore necessary to have a theoretically grounded measure that focuses first and foremost on the relationship between partners and the relationship between parents and children, where phubbing is one of the phenomena that could improve the understanding of family relationships.

Response 1: Thanks for your critical comment. We would clarify the rationality of our study from the following aspects. First, the spillover hypothesis proposed that the expression of mood, effect, or behavior generated in one system (e.g., the marital dyad) can spill over and affect another system (e.g., the parent-child dyad) [1]. For instance, marital relationship quality has been shown to have a significant impact on parent-child relationship quality [2-4]. Previous research has demonstrated that partner phubbing had adverse influences on marital relationship quality [5-8]. Based on the spillover hypothesis [1], the negative effect of partner phubbing on marital relationship quality may further spill over to parent-adolescent relationship quality.

Second, this study assessed partner phubbing using the 9-item phubbing scale developed by Roberts and David [9]. This scale is the most widely used and authoritative instrument for measuring phubbing. It reflects the daily perceived phubbing behavior from the partner but does not measure the specific reasons and motives behind phubbing.

Furthermore, according to your suggestion, we added a future direction that phubbing can be measured in more specific situations where the relationship between partners and the relationship between parents and children are taken into consideration.

References:

  1. Erel, O.; Burman, B. Interrelatedness of marital relations and parent-child relations: A meta-analytic review. Bull.1995, 118, 108–132.
  2. Li, C.; Jiang, S.; Fan, X.; Zhang, Q. Exploring the impact of marital relationship on the mental health of children: Does parent–child relationship matter? Health Psychol.2020, 25, e16691680.
  3. Peltz, J.S.; Rogge, R.D.; Sturge-Apple, M.L. Transactions within the family: Coparenting mediates associations between parents’relationship satisfaction and the parent–child relationship. Fam. Psychol. 2018, 32, 553–564.
  4. Peterson, J. L; Zill, N. Marital disruption, parent-child relationships, and behavior problems in children[J]. Marriage Fam. 1986, 295-307.
  5. Wang, X.; Xie, X.; Wang, Y.; Wang, P.; Lei, L. Partner phubbing and depression among married Chinese adults: The roles of relationship satisfaction and relationship length. Individ. Differ. 2017, 110, 12–17.
  6. Chen, Z.; Gong, Y.; Xie, J. From phubee to phubber: The transmission of phone snubbing behavior between marital partners. Technol. People.2022, 35, 1493–1510.
  7. Wang, X.; Zhao, F.; Lei, L. Partner phubbing and relationship satisfaction: Self-esteem and marital status as moderators. Psychol.2021, 40, 3365–3375.
  8. Beukeboom, C.J.; Pollmann, M. Partner phubbing: Why using your phone during interactions with your partner can be detrimental for your relationship. Comput.Hum. 2021, 124, e106932.
  9. Roberts, J.A.; David, M.E. My life has become a major distraction from my cell phone: Partner phubbing and relationship satisfaction among romantic partners. Hum. Behav.2016, 54, 134–141.

Reviewer 4 Report

The paper deepens the impact of phubbing in couple relationships on the relationship of parents with their children. In my opinion the study is very interesting and well structured and presented.

My only suggestion is to better describe, in section 2.2. Measures, the instrument Parent-Adolescent Relationship Quality, to understand what kind of dimension of relationship between parent and adolescent are considered. 

Author Response

Point 1: My only suggestion is to better describe, in section 2.2. Measures, the instrument Parent-Adolescent Relationship Quality, to understand what kind of dimension of relationship between parent and adolescent are considered.

Response 1: Thank you so much for your affirmation of our work and your constructive suggestion. We sincerely appreciate your help in revising and improving this paper. As for your concern about the dimensions of relationship between parent and adolescent, in the current version, we supplemented the definitions of the four dimensions and present an example item for each dimension. Specifically, the dimension “understanding and communication” refers to that parents fully understand and support their children, and they can communicate unimpeded; an example item is “I know what my child likes and dislikes.” The dimension “excoriation and controlling” refers to the parents’ attempts to control their children by their own subjective will, and this dimension is reversely scored; an example item is “My conversations with my child are imperative or interrogative.” The dimension “liking and esteem” refers to that parents have affection and respect for their children; an example item is “I like being with my child.” The dimension “growth and tolerance” refers to that parents care about their children’s growth and have a tolerant attitude towards their children’s words and deeds during their growth; an example item is “I am tolerant of my child’s failure in trying something.” Detailed information can be found on pages 5-6 lines 238-248 in red.

Round 2

Reviewer 1 Report

thanks for the revised article